# Current Progress in Uterus Transplantation Research in Asia

**DOI:** 10.3390/jcm8020245

**Published:** 2019-02-14

**Authors:** Iori Kisu, Yu Liu, Gaowen Chen, Min Jong Song, Cherry Yin-Yi Chang, Tan Hak Koon, Kouji Banno, Daisuke Aoki

**Affiliations:** 1Department of Obstetrics and Gynecology, Keio University School of Medicine, Tokyo 1608582, Japan; kbanno@keio.jp (K.B.); aoki@z7.keio.jp (D.A.); 2Department of Gynecology, Obstetrics and Gynecology Hospital of Fudan University, Shanghai 200011, China; liuyu160046@126.com; 3Department of Obstetrics and Gynecology, Zhujiang Hospital, Southern Medical University, Guangzhou 510282, China; cgw2012@163.com; 4Department of Obstetrics and Gynecology, Daejeon St. Mary’s hospital, Daejeon 34943, Korea; bitsugar@catholic.ac.kr; 5Department of Obstetrics and Gynecology, China Medical University Hospital, Taichung 40447, Taiwan; d4754@mail.cmuh.org.tw; 6Department of Obstetrics and Gynecology, School of Medicine, China Medical University, Taichung 40447, Taiwan; 7Department of Obstetrics and Gynecology, Singapore General Hospital, Singapore 169608, Singapore; tan.hak.koon@singhealth.com.sg

**Keywords:** uterus transplantation, uterine factor infertility, Mayer-Rokitansky-Küster-Hauser syndrome

## Abstract

Uterus transplantation (UTx) is now a possible approach for women with absolute uterine factor infertility to deliver a child, following the first successful delivery by Brännström et al. in Sweden in September 2014. This remarkable achievement attracted major attention worldwide and caused many countries to prepare for UTx, including countries in Asia. To date, three groups have performed UTx in humans in Asia, and many others are aiming for the clinical application of UTx with accumulation of basic experimental data. Therefore, it is likely that UTx will expand rapidly in Asia in the near future, although this will depend on ethical, social and religious views in each country. With this background, herein we summarize the current progress of UTx in East, Southeast and South Asia, with the purpose of increasing understanding of the current status of basic and clinical UTx research in each country and sharing progress and knowledge to ensure future development of UTx research in Asia.

## 1. Introduction

Uterus transplantation (UTx) is now an alternative to gestational surrogacy and adoption for women with absolute uterine factor infertility (AUFI) to have a child. Brännström et al. achieved the first human delivery after UTx in 2014 in Sweden [1] and this great achievement attracted attention worldwide and led many countries to prepare for UTx. As of October 2018, more than 50 UTx procedures had been performed in Saudi Arabia, Turkey, Sweden, China, USA, Czech, Brazil, Germany, Serbia, India and Lebanon, and a total of 13 babies were born after these procedures in Sweden, USA, Brazil, Serbia and India. Many other countries are now aiming for clinical application of UTx, including France, UK, Belgium, Spain, Italy, Japan, South Korea, Singapore, Taiwan, Australia, Russia, Egypt, Mexico, Colombia and Argentina. Thus, UTx is rapidly spreading worldwide and is starting to be recognized as a new assisted reproductive technology (ART) and transplantation technique, although it is still at the experimental stage and has many medical, technical and ethical issues to be resolved [2].

UTx has mainly been performed in European countries, in many of which surrogacy is not permitted, but research and clinical application of UTx has recently also spread in Asia. Therefore, it is important to understand the current status of basic and clinical UTx research and share knowledge to ensure future progress of UTx research in Asia. With this background, we herein summarize the current progress of UTx in East, Southeast and South Asia based on international and domestic publications, press releases and personal communications.

## 2. Uterus Transplantation (UTx) Research Groups in Asia

To the best of our knowledge, there are 13 groups currently working on UTx research in Asia: seven in China, two in Japan, and one each in India, Korea, Singapore and Taiwan (Figure 1). A summary of groups performing UTx research is given in Table 1.

## 3. Basic UTx Research

A relatively large number of basic UTx studies in large animals, including non-human primates, have been performed in Asia, compared to other areas of the world (Table 2). Among them, Kisu’s group from Keio University Hospital in Tokyo, Japan has accumulated a large UTx data archive in large animal basic studies. There are an estimated 50,000–60,000 women of reproductive age with AUFI in Japan [3] and gestational surrogacy is forbidden in Japan [4]. Basic research on UTx in animals has been performed for approximately 10 years based on the view that extensive training and preparation through preclinical experiments, as also done by the Swedish group for over a decade, are likely to be necessary for successful human UTx.

UTx research in a non-human primate (cynomolgus macaque) was launched in 2009 because the guidelines of the International Federation of Gynecology and Obstetrics (FIGO) indicate that adequate studies in large animals, including primates, should be conducted before clinical application of UTx in humans [5]. Experiments in more than 80 cynomolgus macaques have now been performed, since these animals are anatomically and physiologically similar to humans, but are about the same size as human newborns (approx. 3 kg); therefore, a delicate surgical technique is required. A large amount of data has been accumulated, as described in a previous review [6], including examination of uterine blood flow [7,8,9], surgical procedures for autologous and allogeneic UTx [9,10,11,12,13,14], methods for organ perfusion in a deceased donor model [15], immunological response and rejection [16], and ischemia/reperfusion injury [17,18]. The major achievements include the first natural pregnancy and delivery after autologous UTx in primates and the first pregnancy after allogeneic UTx in a non-human primate experimental model in 2012 [11] and 2018 (unpublished data), respectively. Another Japanese group, led by Yamamuro at Nagoya Daini Red Cross Hospital, is currently preparing for launch of UTx animal experiments, but none have been performed to date.

Most UTx groups in Asia are based in China, the reason for which might be under the background that domestic animal studies of UTx began earlier in Asia. At the start of the century, three Chinese groups began to explore animal models of UTx at almost the same time, and also at about the time UTx research began in Sweden. However, results of the studies in China were published in domestic journals, and thus failed to achieve international attention until Wei’s group (Xijing Hospital, X’ian) performed the first human UTx in China [19]. UTx studies in rat, rabbit, dog, pig, cynomolgus monkey and rhesus monkey have produced important information on vascular harvesting, uterine perfusion, ischemia injury, uterus preservation, and characteristics of graft rejection after allogeneic UTx. Live births have been achieved after syngeneic UTx in rat [20] and autologous UTx in dog [21] models. Research findings from the seven Chinese groups (Table 1) are summarized below. 

Wei’s group at Xijing Hospital (Fourth Military Medical University, Xi′an) performed allogeneic UTx in 10 sheep as a pilot study prior to human application [22]. The advantage of using sheep is that the body size and pelvic vascular anatomy are similar to those in young women, although the uterus is bicornuate. Techniques for orthotopic and allogeneic UTx and subsequent immunosuppressive protocols were developed and evaluated in sheep. Two of the 10 transplanted uteri showed signs of estrus after transplantation. 

Based on the FIGO recommendations [5], since 2010 Wang and Yu’s group at Zhujiang Hospital (Southern Medical University, Guangzhou) have prepared for UTx in non-human primates through experiments in the cynomolgus macaque model. In four initial autologous UTx procedures, one animal died intraoperatively due to venous hemorrhage. At 30–45 days after UTx, second-look laparotomy was performed in the surviving animals, but no signs of cyclicity and menstruation were found. The internal genitalia and pelvic cavity had severe adhesions and the utero-tubo-ovarian system was atrophic. This team subsequently attempted two allogeneic UTxs, but with poor postoperative results. It is regrettable that this initial outcome has yet to be published (Table 2).

After accumulating knowledge and skills in previous work, Wang and Chen’s group (Zhujiang Hospital, Guangzhou) further optimized surgical strategies in 6 autologous [23] and 4 allogeneic UTxs in cynomolgus macaques and two allogeneic UTxs in rhesus macaques (unpublished data). Menstruation recovered following two of the autologous UTxs, but with no subsequent pregnancy. In the allogeneic UTx models, no menstruation was established, but cyclicity resumed in one rhesus macaque postoperatively. In these studies, different surgical protocols were used for venous anastomosis in the primate models, and finally anastomosis of utero-ovarian veins, rather than uterine veins, was chosen to reestablish drainage of the transplanted uterus. All animals, including those resuming menstruation, underwent second-look laparotomy or autopsy, in which moderate to severe pelvic adhesion over the internal genitalia was found in all cases. Therefore, IVF-ET (in-vitro fertilization-embryo transfer) was abandoned in two menstruating cynomolgus macaques after autologous UTx. Despite the poor results of allogeneic UTx, performance of UTx in a primate model provides ideal training for surgeons in vascular dissection and anastomosis. This then permits effective procurement of the uterus in brain-dead human donors. 

There are two UTx groups in Shanghai. Hua and Liu’s group at the Obstetrics and Gynecology Hospital (Fudan University) has done extensive research on preservation of female fertility [24,25]. This group has treated a significant number of patients with genital malformation who wanted to have their own child, and are committed to extending UTx to the clinical stage [26,27]. The New Zealand white rabbit was chosen as the primary animal model for UTx because the use of primates as animal models for development of surgical methods raises ethical and economic concerns. In addition, UTx using rabbits had not been performed in China. Two allogeneic UTxs were performed in rabbits and short-term survival occurred in both cases. The surgical protocol and immunosuppressive therapy were learned from a British group [28,29]. Due to the small complicated branch of the rabbit uterus blood vessels, a modified perfusate containing nitroglycerin had to be used to achieve adequate uterine perfusion. The two allogeneic UTx procedures were completed, but long-term survival was not achieved and histopathology showed severe uterus rejection, with immunosuppression with tacrolimus and hydrocortisone seemingly insufficient (unpublished data). Further exploration of these results is required. Thereafter, this group observed UTx in a cynomolgus macaque conducted by Kisu’s group in July 2018, and studied the allogeneic UTx cynomolgus macaque model with regard to aortic-caval macrovascular patch harvesting, MHC gene typing, and the immunosuppression protocol, which are all important for the success of the procedure. Studies using the cynomolgus macaque are currently being established in Hua and Liu’s group. 

Wan’s group from Shanghai First Maternity and Infant Hospital (Tongji University) has done a lot of work in establishing UTx models in rats, and examining rejection patterns and immunosuppressive therapy for allogeneic UTx in rats [30,31,32,33]. The most important study used 20 female Wistar rats aged 8–10 weeks as donors and recipients, and the recipients then mated spontaneously with male Wistar rats for 12 weeks. Ten UTx procedures in syngeneic rats succeeded in restoring fertility in 50% of the animals, with live birth of 20 rats (20). No recent results have been reported by this group.

In 2001, when the first UTx research started in China, Chen’s group from the Reproductive Hospital (Shandong University, Jinan) explored orthotopic uterus autologous UTx models in 10 beagle bitches, of which 6 survived surgery with a 66.7% (4/6) graft survival rate, and 2 achieved long term survival, including one with hormone-induced estrus that delivered 3 live puppies vaginally [21]. Thereafter, another group led by Zhang from The Jilin University Second Hospital in Jilin performed allogeneic UTx on mongrel dogs. This study revealed that the acceptable warm ischemia time for UTx in dogs is up to 60 min, and 5 dogs (62.5%) survived the surgery, but none survived for >96 h [34]. The main reasons for the poor survival were anastomotic bleeding and infection. Nowadays, dogs are seldom used as a UTx model.

Ten years later, Yao’s group from the Chinese People’s Liberation Army General Hospital in Beijing performed autologous UTx in 5 pigs, of which 3 survived surgery, including one with long-term survival of >6 months. This group then performed allogeneic UTx in a rhesus macaque that survived for 28 days after surgery, with a probable cause of death of organ failure due to anorexia and chronic anemia, without pathological evidence of rejection [35]. Recently, Yao’s group reported allogeneic UTx in 5 pigs, with 100% surgical success and 80% long-term survival. One pig resumed temporary estrus, but gestation was not achieved after artificial embryo transfer [36].

In Korea, Song’s group from the Catholic University Medical Center (Catholic University, Seoul) has accumulated experimental and clinical data in collaboration with Brännström et al. in Sweden. This group conducted autologous UTx in 10 rats and 3 domestic pigs, and achieved successful organ perfusion after autologous UTx in these animals. The Korean group has also developed a bioengineered uterus in rats for future UTx. The Korean and Swedish groups have established a protocol for decellularization of the rat uterus and are now examining bioscaffold rejection. Song’s group also observed UTx in cynomolgus macaques conducted by Kisu’s group in Japan in July 2018 and is ready to perform preclinical studies such as fresh cadeveric uterine dissection.

In Singapore, initial interest in exploring the feasibility of UTx led to informal discussions in 2012 between the Departments of Obstetrics and Gynaecology and Plastic and Reconstructive Surgery at Singapore General Hospital, the largest tertiary teaching hospital in Singapore. In 2013, a Uterine Transplantation Group (Tan’s group from Singapore General Hospital) was formed, comprising gynaecologists, vascular surgeons, plastic surgeons, transplant physicians, high-risk obstetricians, and psychologists. One autologous UTx, followed by two allogeneic UTxs, in cynomolgus macaques were performed (one autologous and one allogeneic UTx under the supervision of Dr. Kisu) and uterine procurement and reimplantation were achieved. The allogeneic UTx did not show evidence of hyperacute rejection for up to 6 h postoperatively with the immunosuppression regime used. One autologous and one allogeneic UTx in sheep were then conducted in Gothenburg, Sweden, under the supervision of Dr. Brännström and his group, and these resulted in two successful sheep uterine procurements and transplantation. 

In Taiwan, Lin’s group from China Medical University Hospital in Taichung have set up the team for the uterine transplantation and performed three autologus UTx in sheep models. In these three sheep, two were confirmed functional uteri after surgery. Gestational surrogacy is forbidden in Taiwan, and researchers need to provide an animal model report to prove sufficient training and preparation based on preclinical experiments prior to clinical application.

## 4. Preclinical Research

Preclinical studies using human cadavers or deceased multi-organ donors are likely to be particularly important before clinical application, especially for surgical training (Table 3). Kisu’s group in Tokyo has performed four fresh cadaveric uterus dissections and transplantations to simulate UTx in humans, two of which were conducted in collaboration with the Singaporean group at the end of 2015. These procedures resulted in two completely dissected uterus specimens, and led to technical advancements in the surgical procedure for dissection of vessels surrounding the uterus and retrieval and reimplantation of the uterus. Kisu’s group and Song’s group in Seoul also observed two human UTx procedures with robot-assisted donor surgery performed in Sweden in April 2018 to learn surgical skills and details of postoperative management.

Since 2015, Wang and Chen’s group in Guangzhou, which has conducted most preclinical studies in Asia, performed uterine retrieval from a brain-dead donor (URBD) in 8 cases (partial data published in a Chinese journal) [37], with the aim of exploring standard procurement techniques and assessing the feasibility of perfusion methods with different protocols. Using different surgical strategies of radical hysterectomy and trachelectomy, it was found that radical en-bloc resection of parametrial tissue should be performed, instead of exposure of the uterine artery, vein and ureter, in URBD to avoid vascular rupture. Procurement of the uterus synchronized with liver retrieval was also achieved in the third and fourth case. Interestingly, this group plans to evaluate the outcome of various organ perfusion solutions at 4 °C using heparinized physiological saline, histidine-tryptophan-ketoglutarate (HTK), UW and Celsior. The group has also screened more than 280 women with AUFI and ultimately selected 11 for human UTx, including one who has already undergone UTx (unpublished data). In another study, Chen’s group investigated the safety of cold extracorporeal perfusion with HTK and ischemic storage of the human uterus, and found that human uterine myometrial tissue can tolerate cold ischemia for at least 6 h when stored in HTK solution after a short time of extracorporeal perfusion [38]. Updates on clinical registration for human UTx are awaited from this group.

## 5. UTx in Humans

Clinical applications of UTx in Asia have only been achieved in China and India, although there are many groups in Asia that have examined animal models of UTx in preparation for clinical application (Table 3). Of the 3 UTx procedures in humans in China to date, two were carried out by this group, and the other by Wang and Chen’s group. Wei’s group in Xi’an performed the first robot-assisted UTx worldwide in November 2015, in which a woman with Mayer-Rokitansky-Küster-Hauser (MRKH) syndrome received a uterus from her mother, and had recovered periodic menstruation without rejection at one year after surgery under the standard maintenance immunosuppression (tacrolimus, steroids, and mycophenolate mofetil (MMF)) [22]. Donor surgery is regarded as highly invasive, and less invasive donor procedures are needed. Therefore, this first trial of robot-assisted donor surgery emphasized the benefits of minimally invasive methods for procurement of the uterus. Also, a new operative procedure was developed for using the ovarian vein as a drainage vein instead of the uterine vein, leading to a reduced operative time and bleeding volume. Consequently, use of robotic surgery and the ovarian vein are likely to reduce risks in living donor surgery for UTx. 

Wang and Chen’s group in Guangzhou performed human UTx from mother to daughter with MRKH syndrome using laparoscopic donor surgery in February 2017 (unpublished data). Vascular grafts harvested with the uterus included the hypogastric arteries, uterine vein and ovarian veins. Uterine veins were anastomosed to external iliac veins after lengthening using the great saphenous veins of the donor. Standard maintenance immunosuppressants were used after surgery. Unfortunately, the transplanted uterus had to be removed on postoperative day 30 due to acute left uterine vein thrombosis caused by the long vascular graft.

Puntambekar’s group from the Galaxy CARE Laparoscopy Institute (Pune, India) performed two UTxs with laparoscopic live-donor uterus retrieval on two successive days in May 2017 [39,40]. The recipients were patients with MRKH syndrome and Asherman syndrome, respectively, and their mothers were the donors. Vessels were harvested laparoscopically in both donors, and the uterus with vascular pedicles was retrieved via a small abdominal incision to prevent injury to the harvested vessels. The operative time for the donor surgery was 4 h and blood loss was approximately 100 mL in both donors, indicating a reduced operative time and bleeding volume. Both recipients started regular menstruation after UTx under the standard maintenance immunosuppressants. In a press release [41], it was announced that one patient had delivered a baby in October 2018, as the first baby born after UTx in Asia and after laparoscopic donor surgery. This outcome shows that laparoscopic donor surgery is feasible and has the advantages of a minimally invasive technique, which is likely to motivate the performance of minimally invasive UTx. 

## 6. Future Directions in Asia

UTx in humans has been conducted worldwide, but more studies are required before UTx can be considered to be a part of general clinical practice. There are many medical, ethical, social and religious issues to be resolved, and the backgrounds to these issues vary among countries in Asia. The ethical aspects of UTx require a thorough discussion in the context of each background.

In Japan, UTx has yet to reach the point of clinical application despite accumulation of a lot of data in animal models. Clinical practice requires not only technology and knowledge acquisition in basic experiments, but also discussion in institutions and academic societies, and consensus-building in the public. To address ethical and social problems, since 2012 more than 10 meetings with academics and specialists in obstetrics and gynecology, surgery, ethics, philosophy, clinical psychology and nursing, and with a representative group of patients with infertility, have been held to discuss clinical application of UTx. In 2014, the Japan Society for Uterus Transplantation (JSUT) was established because discussions beyond the internal debate and conveyance of information to the public were needed for further progress [42]. The JSUT informs the public about UTx technology; addresses medical, social and ethical perspectives on UTx; and discusses clinical application of UTx in Japan. In 2018, Kisu’s group from Keio University Hospital in Tokyo requested that the Japan Society of Obstetrics and Gynecology and the Japan Society for Transplantation establish a consensus on clinical application of UTx in Japan. The Japanese Association of Medical Science plans to set up a joint committee of multidisciplinary experts. Due to concern regarding the burden of the donor in the surgical procedure and the Japanese conservative view of innovative techniques with limited evidence, discussion on clinical application in Japan is unlikely to proceed smoothly. 

Gestational surrogacy is forbidden in China, and although adoption can lead to legal motherhood, only UTx gives the mother legal, biological and gestational motherhood. Clinical application of UTx in China had been attempted since 2015, with three UTxs performed from living donors (one at Zhujiang Hospital and two at Xijing Hospital). No pregnancy has been reported and detailed data will be published in the near future. Progress on the ethics of human UTx in China seems slow (unpublished data), despite development of various UTx models in different animals and preclinical studies, as discussed above. A national internet-based survey is suggested regarding the attitudes of the public and gynecologists towards UTx. It is recognized that recipients should be adequately informed of the surgical qualifications of the group, the risks of transplantation surgery and immunosuppressive treatment; possible pregnancy complications under immunosuppression; and the potential need for a subsequent hysterectomy.

From social and legal perspectives, UTx seems to have gained more acceptance in China compared with Japan and other countries in Asia. Thus, a few Chinese researchers have received government grants to conduct basic and clinical studies of UTx. However, since the senior researchers prefer to publish their outcomes in Chinese journals in the parent language, most of the results are unknown outside China. It would be positive for the Chinese groups to establish an academic society for UTx, as done by Kisu’s group, but based on Chinese cultural traditions and socioeconomic conditions. Regular academic meetings will promote mutual learning, raise public awareness of UTx, and allow exchange of the latest developments on UTx at the international level. The first case worldwide of live birth following UTx from a deceased donor in Brazil is inspiring [43], and recent conference data shows that more than 600,000 people have registered to become volunteer organ donors in China, with nearly 20,000 having donated organs [44]. Thus, the UTx groups can work with the China Organ Transplant Response System (COTRS) and Red Cross office to address the ethics of uterine transplants.

In Korea, it is estimated that there are more than 10,000 women of reproductive age with AUFI based on the prevalence (there are no published statistical data) and gestational surrogacy is forbidden. The team at Catholic University Medical Center set up a UTx task force in 2014 to conduct preclinical and animal studies, as preparation for human UTx. This task force plans to establish a multidisciplinary expert meeting, including gynecologists, transplant surgeons, ethical scholars, priests and clinical psychologists, to address medical, ethical and social problems; and is also ready to launch the Korean Society of Uterus Transplantation to inform the public about UTx and promote collaboration among academic groups. 

In Singapore, the UTx group has successfully obtained approval from the Bioethics and Centralized IRB (Institutional Review Board). However, Singapore recently enacted a Human Biomedical Research Act, and the group has submitted their UTx research project to the Ministry of Health through the Tissue and Research Application System. Permission from the Ministry is likely. In Taiwan, completion of practice in animals and sharing the data with the Ministry of Health and Welfare is required for approval of a new technique. The Taiwan Association of Obstetrics and Gynecology has welcomed progress in UTx but the institution or hospital which plans to perform the procedure need to obtain approval of the practice from the government. Regarding Puntambekar’s group in India, which is the only one in Asia to succeed in delivery after UTx, further procedures and deliveries are likely, along with sharing of accumulated experience and techniques, especially in laparoscopic donor surgery, in Asia and worldwide.

## 7. Conclusions

Despite still being in the experimental stage, UTx is now a possible approach for women with AUFI to deliver a child. Clinical application of UTx has spread rapidly to many countries, including in Asia, following the first successful delivery after UTx in Sweden. This has led several groups to conduct UTx and many to prepare for clinical trials. However, we notice that several medical centers around the globe are declaring a potential schedule for human UTx without adequate preparatory studies. Establishment of UTx as a new therapy requires accumulation of clinical study data, thorough training and testing in animal models, and international collaboration and information sharing. Moreover, differences in social, ethical and religious backgrounds among countries make it important to examine whether the clinical use of UTx is desirable in each society. It is likely that UTx will expand rapidly in Asia in countries with differences in these backgrounds. Therefore, we believe that it is beneficial to understand the current situation of UTx in each country, to share progress and knowledge, and to collaborate with each other internationally to ensure the future development of UTx in Asia. 

## Figures and Tables

**Figure 1 jcm-08-00245-f001:**
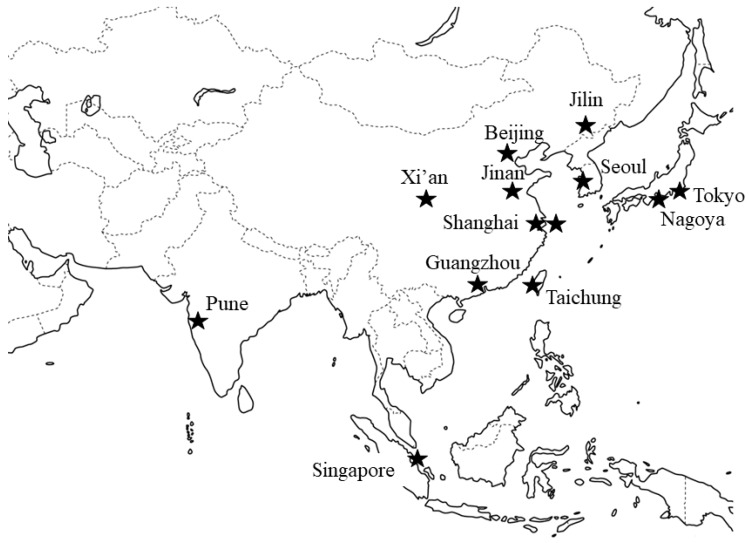
Current groups for UTx (uterus transplantation) research in Asia.

**Table 1 jcm-08-00245-t001:** Groups performing UTx (uterus transplantation) research in Asia.

Country	City	Team Leader	University/Hospital
China	Xi′an	Wei L	Fourth Military Medical University/Xijing Hospital
	Guangzhou	Wang YF/Chen GW	Southern Medical University/Zhujiang Hospital
	Shanghai	Hua KQ/Liu Y	Fudan University/Obstetrics and Gynecology Hospital of Fudan University
		Wan XP	Tongji University/Shanghai First Maternity and Infant Hospital
	Jinan	Chen ZJ	Shandong University/Reproductive Hospital Affiliated with Shandong University
	Beijing	Yao YQ	Chinese People's Liberation Army General Hospital
	Jilin	Zhang WY	Jilin University/The Jilin University Second Hospital
India	Pune	Puntambekar S	Galaxy CARE Laparoscopy Institute
Japan	Tokyo	Kisu I	Keio University/Keio University Hospital
	Nagoya	Yamamuro O	Nagoya Daini Red Cross Hospital
Korea	Seoul	Song MJ	Catholic University/Daejeon St. Mary's Hospital
Singapore	Singapore	Tan HK	Singapore General Hospital
Taiwan	Taichung	Lin WC	China Medical University/China Medical University Hospital

**Table 2 jcm-08-00245-t002:** Basic UTx research in Asia.

Country	Team Leader	Animals	Study	Number	Evaluation	Main Outcome
China	Wei L	Sheep	Allogeneic UTx	10	UPT	Showing signs of estrus after allogeneic UTx in 2 of 10 transplanted uteri
	Wang YF/Yu P	Cynomolgus macaque	Autologous UTx	4	UPT	Out of four animals, one died intraoperatively due to venous hemorrhage. Surviving animals underwent second-look laparotomy within 30–45 days
			Allogeneic UTx	2	UPT	One animal died on postoperative day 2 because of abdominal infection. One survived without cyclicity or menstruation
	Wang YF/Chen GW	Cynomolgus macaque	Autologous UTx	6	UPT	Recovery of menstruation (2 of 6) but no pregnancy
			Allogeneic UTx	4	UPT	No menstruation after alloUTx
		Rhesus macaque	Allogeneic UTx	2	UPT	No menstruation after alloUTx but cyclicity resumed in one animal
	Hua KQ/Liu Y	Rabbit	Allogeneic UTx	2	UPT	Successful allo UTx with short-term survival
			Perfusion	4	Uterine harvesting and perfusion	Successful uterine harvesting and perfusion (3/4)
		Cynomolgus macaque	Allogeneic UTx		N/A (ongoing)	N/A
	Wan XP	Rat	Syngeneic UTx	10	UPT	Restored fertility (50%) with live birth to 20 rats
			Syngeneic UTx	30	UPT	Recipient survival 60% (18/30), graft survival 83.3% (15/18)
			Syngeneic UTx	60	UPT	Recipient survival 88.3% (53/60), graft survival 88.7% (47/53) Establishing UTx models in rats
			Allogeneic UTx	30	Rejection patterns of uterine transplant	Recipient survival 100%, acute rejection of allogeneic UTx in rats involves IFN-γ
			Allogeneic UTx	36	Immunosuppressive therapy	Recipient survival 100%, Cyclosporine A has significant therapeutic effect on allograft rejection after UTx in rats
	Chen ZJ	Dog	Autologous UTx	10	UPT	Recipient survival rate 60% (6/10), graft survival 66.7% (4/6) with 2 long-term survival, one hormone-induced estrus and vaginal delivery of 3 live puppies
	Yao YQ	Pig	Autologous UTx	5	UPT	Three survived from surgery, one long-term survival (>6 mo)
			Allogeneic UTx	5	UPT	Surgical success 100%, long-term survival 80%, one had temporary estrus resumed
		Rhesus macaque	Allogeneic UTx	1	UPT	Death due to chronic failure 28 days after surgery
	Zhang WY	Dog	Allogeneic UTx	8	UPT	Five survived from surgery, none survived more than 96 h
			Ischemia	10	Allowable warm ischemic time	Acceptable warm ischemia time for UTx in dogs is up to 60 min
India	Puntambekar S	Unknown	Unknown		Unknown	Unknown
Japan	Kisu I	Cynomolgus macaque	Autologous UTx	9	UPT	Pregnancy and delivery after auto-UTx
			Allogeneic UTx	14	UPT	Pregnancy after allo-UTx
			Uterine blood flow	1	Uterine blood flow using ICG fluorescence	ICG fluorescence imaging is useful for evaluation of uterine blood flows
			Procurement	22	Minimally invasive donor surgery	Using ovarian veins reduce risks for donors
			Ischemia	18	Warm ischemia/reperfusion injury	Allowable warm ischemic time >4 h
			Perfusion	3	Organ perfusion in brain dead donor models	Perfusion via femoral artery and/or external iliac artery could be useful
			Uterine rejection	3	Clinical features associated with irreversible rejection	Increases in WBC, LDH and CRP and uterine shrinkage after transient swelling in irreversible rejection
	Yamamuro O	N/A	N/A		N/A	N/A
Korea	Song MJ	Rat	Autologous UTx	10	UPT	Successful perfusion after autologous UTx
			Bioengineering	60	Bioengineered uterus using a bioscaffold	Establishment of protocol for decellularization of rat uterus
		Pig	Autologous UTx	3	UPT	Successful perfusion after autologous UTx
Singapore	Tan HK	Sheep	Autologous UTx	1	UPT	Successful UPT
			Allogeneic UTx	1	UPT	Successful UPT
		Cynomolgus macaque	Autologous UTx	1	UPT	Successful UPT
			Allogeneic UTx	2	UPT	Any evidence of hyperacute rejection for up to 6 h postoperatively
Taiwan	Lin WC	Sheep	Allogeneic UTx	3	UPT	Functional uterus after allogeneic UTx

* UPT: uterine procurement and transplantation, ICG: indocyanine green, WBC: white blood cell, LDH: lactate dehydrogenase, CRP: C-reactive protein.

**Table 3 jcm-08-00245-t003:** Preclinical studies and human UTx research in Asia.

Country	Team Leader	Performance of UTx	Pre-Clinical Study or Human UTx	Main Outcome	Approval of Ethical Committee
China	Wei L	Yes	First human robot-assisted UTx	Successful uterine procurement and recovery of uterine function	Done
	Wang YF/Chen GW	Yes	Cold ischemic preservation of uterine tissue, uteri retrieval (8 cases), organ perfusion and ischemic study (4 cases) * All studies performed from brain-dead donors	Show that uterine tissues are nearly normal in 2.5 h after perfusion Confirm different perfusion methods via external iliac artery or abdominal aorta.	Done
			UTx with living donor by laparoscopic donor surgery	Removal of transplanted uterus 30 days after surgery (acute left uterine vein thrombosis)	
	Hua KQ/Liu Y	No	No	-	Ongoing
	Wan XP	No	No	-	No
	Chen ZJ	No	Allowable cold ischemic time in human uterus with HTK (histidine-tryptophan-ketoglutarate) as perfusate (4cases)	Human uterine myometrial tissue tolerates cold ischemia for at least 6 h in HTK solution	Done
	Yao YQ	No	No	-	No
	Zhang WY	No	No	-	No
India	Puntambekar S	Yes	UTx with living donor by laparoscopic donor surgery	First successful delivery in Asia	Done
Japan	Kisu I	No	Cadeveric uterine dissection (4 cases) * 2 cases performed with Singaporean team	Successful dissection of vessels surrounding the uterus and retrieval of the uterus	No
	Yamamuro O	No	No	-	No
Korea	Song MJ	No	No	-	No
Singapore	Tan HK	No	Cadeveric uterine dissection (3 cases) * 2 cases performed with Japanese team	Successful dissection of vessels surrounding the uterus and retrieval of the uterus	Done
Taiwan	Lin WC	No	No	-	No

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
