# Peer review of "Current Progress in Uterus Transplantation Research in Asia"

_jcm, 2019, doi:10.3390/jcm8020245_

Round 1
Reviewer 1 Report
The study was well articulated but would benefit from some necessary inclusions. Author should proffer possible reasons why most UTx research groups are based in China.
The section on ‘basic’ UTx research contains a number of clinical studies as well. Authors should address in order to ensure a smooth storyline.
Authors should include more details on baby survival after birth from these procedures, postnatal complications if any that may have arisen. Specifics on time at which UTx has been shown to most effective and if drug therapy is required during the maintenance period besides antibiotics.
Author Response
Response to Reviewer #1
We are grateful to Reviewer #1 for the critical comments and useful suggestions that have helped us to improve our paper considerably. As indicated in the responses that follow, we have taken all these comments and suggestions into account in the revised version of the paper.
Comments: The study was well articulated but would benefit from some necessary inclusions. Author should proffer possible reasons why most UTx research groups are based in China.
Response: We are grateful for the positive comments on our work. We have added the description for this comment in the text in the section of “Basic UTx research”.
Comments: The section on ‘basic’ UTx research contains a number of clinical studies as well. Authors should address in order to ensure a smooth storyline.
Response: We have moved the description on clinical studies in the section of ‘basic UTx research’ to the section of ‘UTx in humans’. The description of ‘The Korean group participated in human UTx performed in Sweden’ in the section of ‘basic UTx research’ has been deleted.
Comments: Authors should include more details on baby survival after birth from these procedures, postnatal complications if any that may have arisen. Specifics on time at which UTx has been shown to most effective and if drug therapy is required during the maintenance period besides antibiotics.
Response: We cannot contact the Indian team which succeeded in delivery after UTx in human and the details of the follow-up of the babies after UTx are not mentioned in published articles. Therefore, we cannot describe them. Drug therapy including immunosuppressants, antibiotics, antiviral drugs and antithrombotic prophylaxis are of course needed after general organ transplantation including UTx, but we believe the details of these drugs in each team are not important and it is not appropriate to describe them in this review. We hope that the reviewer will accept these answers.
Reviewer 2 Report
The manuscript reviews the progression of pre-clinical studies and clinical practices about uterus transplantation in Asia. It is well organized and professionally written.
Is there any difference of drug treatment in three groups of human UTx?
Author Response
Response to Reviewer #2
We are grateful to Reviewer #2 for the critical comment that have helped us to improve our paper considerably. As indicated in the response that follows, we have taken all comments into account in the revised version of the paper.
Comments: The manuscript reviews the progression of pre-clinical studies and clinical practices about uterus transplantation in Asia. It is well organized and professionally written.
Is there any difference of drug treatment in three groups of human UTx?
Response: We are grateful for the positive comments on our work. We have added the description of immunosuppressants in the text in the section of “UTx in human”.